# The Prevalence of Cardiac Diseases in a Contemporary Large Cohort of Dutch Elderly Ankylosing Spondylitis Patients—The CARDAS Study

**DOI:** 10.3390/jcm10215069

**Published:** 2021-10-29

**Authors:** Milad Baniaamam, Sjoerd C. Heslinga, Laura Boekel, Thelma C. Konings, M. Louis Handoko, Otto Kamp, Vokko P. van Halm, Irene E. van der Horst-Bruinsma, Mike T. Nurmohamed

**Affiliations:** 1Reade Rheumatology, Amsterdam Rheumatology and Immunology Center, 1056 AB Amsterdam, The Netherlands; l.boekel@reade.nl (L.B.); m.nurmohamed@reade.nl (M.T.N.); 2Amsterdam Cardiovascular Sciences, Vrije Universiteit, 1081 HV Amsterdam, The Netherlands; 3Department of Rheumatology, Amsterdam UMC, Vrije Universiteit Amsterdam, 1081 HV Amsterdam, The Netherlands; scheslinga@gmail.com (S.C.H.); i.e.vanderhorst@amsterdamumc.nl (I.E.v.d.H.-B.); 4Department of Cardiology, Amsterdam UMC, Vrije Universiteit Amsterdam, 1081 HV Amsterdam, The Netherlands; t.konings@amsterdamumc.nl (T.C.K.); ml.handoko@amsterdamumc.nl (M.L.H.); o.kamp@amsterdamumc.nl (O.K.); v.p.vanhalm@amsterdamumc.nl (V.P.v.H.)

**Keywords:** ankylosing spondylitis, cardiovascular disease, left ventricular function, cardiac conduction disturbances, transthoracic echocardiography

## Abstract

Objectives: The aim of the present study was to determine the prevalence of specific cardiac manifestations, i.e., conduction disorders, valvular disease and diastolic left ventricular (LV) dysfunction, in a large cross-sectional controlled cohort of elderly ankylosing spondylitis (AS) patients. Methods: This cross-sectional study assessed the prevalence of valvular disease, conduction disorders and LV dysfunction in 193 randomly selected AS patients compared with 74 osteoarthritis (OA) controls aged 50–75 years. Patients underwent conventional and tissue Doppler echocardiography in combination with clinical and laboratory assessments. Multivariate regression analyses were performed to compare the odds of mitral valve regurgitation (MVR) and aortic valve regurgitation (AVR) between AS patients and OA controls. Results: The prevalence of diastolic dysfunction was trivial and comparable in AS patients compared to controls (respectively, 4% and 3%) and had no further clinical relevance. In addition, the prevalence of conduction disturbances was similar in both groups, with little clinical relevance, respectively 23% vs. 24%. The prevalence of AVR was significantly higher in AS patients compared to the controls, respectively 23% (9% trace, 12% mild, 1% moderate, 1% severe, 1% prosthesis) vs. 11%, *p* = 0.04. After correcting for age, sex and CV risk factors, AS patients had an odds ratio of 4.5 (95% CI 1.1–13.6) for AVR compared to the controls. In contrast, the prevalence values of MVR were similar and mostly not clinically relevant in AS patients and controls, respectively 36% and 32% and *p* = 0.46. Conclusion: The prevalence of diastolic LV dysfunction and conduction disorders was mostly not clinically relevant, and similar in AS patients and controls. However, AS patients had an up to five times increased odds to develop AVR compared to controls. Therefore, echocardiographic screening of elderly (50–75 years) AS patients should be considered.

## 1. Introduction

Ankylosing spondylitis (AS) is an inflammatory joint disease associated with extra-articular manifestations, including cardiac disease [1,2]. Previous studies have shown an increased mortality in AS patients compared to the general population, with cardiovascular diseases as the leading cause of death [3,4]. Chronic inflammation in AS contributes considerably to this excess cardiovascular risk; besides progression of atherosclerosis, it may lead to structural changes of the heart, thereby causing cardiac disease [5,6]. Indeed, previous studies suggest that AS patients are at increased risk to develop aortic valve regurgitation (AVR), conduction disorders and diastolic left ventricular (LV) dysfunction. Although some of the previous studies suggested that these cardiac manifestations occur more frequently in AS patients, results varied and are altogether inconclusive [7,8,9,10,11,12,13,14,15]. Furthermore, therapeutic treatment in AS has significantly improved in the last decades and this may have altered the CV burden. As recent data regarding the prevalence of cardiac manifestations in AS are lacking, clear guidelines regarding cardiovascular screening of AS patients are still absent. Hence, current cardiovascular risk management guidelines make no distinction between AS patients and healthy persons, and the EULAR only recommends to perform a CVD risk assessment once every 5 years [16]. Therefore, the aim of the present study was to determine the current prevalence of diastolic LV dysfunction (primary objective), cardiac valve regurgitation and conduction disorders (secondary objectives) in a large cohort of AS patients, and compare these results to findings in controls without an inflammatory joint disease, i.e., osteoarthritis (OA) patients. As cardiac disease is more prominent with older age, we assessed the prevalence of cardiac disease in subjects between 50 and 75 years.

## 2. Materials and Methods

### 2.1. Study Population

A cross-sectional controlled study was conducted in randomly selected AS patients and age, sex and smoking status matched controls in a 2:1 ratio. Control patients with OA were selected as they also suffer from joint problems and subsequent mobility issues (physical activity), however without auto-inflammatory characteristics. Subjects were randomly recruited between March 2014 and February 2020 at a large rheumatology outpatient clinic (Reade) in Amsterdam, the Netherlands. Patients were eligible for inclusion if they were between 50 and 75 years of age. AS patients needed to be diagnosed according to the 1984 modified New York [17] criteria and OA controls with active hip-, knee- or poly-osteoarthritis had to be diagnosed by a general practitioner or rheumatologist. Patients with a history of chemotherapy (for malignant disease) were excluded due to the presence of potential cardiotoxicity. All patients provided written consent prior to inclusion in the study. This study was conducted in accordance with the Helsinki Declaration, and the protocol (NL44202.048.13) was approved by the medical ethics committee of the Slotervaart hospital and Reade, Amsterdam, the Netherlands.

### 2.2. Echocardiography

Transthoracic echocardiography (TTE) was performed by certified echo technicians at the European Society of Cardiology (ESC)-certified department of echocardiography of the Amsterdam University medical center, location VUmc, using a Philips ultrasound system (Epiq 7 or IE 33). Furthermore, the echo technicians were not informed about the clinical diagnosis of the subject. To exclude inter-observer variability, all recordings of echocardiographic images and data were assessed afterwards by an experienced cardiologist of the VUmc specialized in echocardiography (T.K.). The cardiologist graded diastolic dysfunction, AVR and MVR. TTE was performed according to the guidelines provided by the American Society of Echocardiography (ASE) and the European Association of Cardiovascular Imaging (EACVI) (see Appendix A) [18]. Furthermore, the severity of AVR and MVR was graded according to the EACVI guidelines [19,20]. Left atrial volume (LA volume, mL) index and left ventricular mass (LVM, g) index were calculated with body surface area (BSA, m^2^) LA volume or LVM/BSA (g/m^2^). The aortic root was measured at sinuses of Valsalva during diastole. Furthermore, aortic root diameter was corrected for BSA according to the Dubois method (aortic root index) [21]. An aortic root index of ≥2.1 cm/m^2^ was considered as aortic root dilatation [22,23].

### 2.3. Electrocardiography

Electrocardiography (ECG) was performed using standard 12-lead ECGs, recorded at a 25 mm/s paper speed. ECGs were analyzed by a single cardiologist (T.K.), who was blinded to the clinical status of all patients.

### 2.4. Disease-Specific Parameters

The following disease-specific parameters were collected. In AS: HLA-B27 status, extra-articular manifestations and disease activity (Bath AS Metrology Index (BASMI), Bath AS Disease Activity Index (BASDAI) and AS Disease Activity Score-C-reactive protein (ASDAS-CRP)). In AS, high disease activity was defined as an ASDAS score of ≥2.1. In OA, disease severity was assessed with the Western Ontario and McMaster Universities Osteoarthritis index (WOMAC) questionnaire [24].

### 2.5. Cardiovascular History and Risk Factor Parameters

Cardiovascular risk factors were assessed including smoking status, body mass index (BMI), hypertension, hypercholesterolemia, diabetes mellitus type II and family history for cardiovascular disease. Furthermore, data history for cardiovascular disease was collected, i.e., angina pectoris, myocardial infarction, congestive heart failure, stroke (cerebrovascular accident (CVA) and/or transient ischemic attack (TIA)), peripheral ischemia and coronary arterial bypass grafting (CABG).

### 2.6. Other Study Parameters

Anthropometric data including length, weight, waist/hip ratio and blood pressure were assessed during physical examination. Demographic data were collected, i.e., age, race, ethnicity and sex. Blood sample measurements (non-fasting) consisted of standard hematological assessment, erythrocyte sedimentation rate (ESR), CRP, triglyceride, total cholesterol, high-density lipoprotein (HDL), low-density lipoprotein (LDL) and HLA-B27 status. Furthermore, medical history, current and historic medication use and disease-specific data (i.e., year of onset, disease duration) were documented.

### 2.7. Definitions

#### 2.7.1. Systolic LV Dysfunction

Systolic LV dysfunction was defined as a left ventricular ejection fraction (LVEF) less than 50%.

#### 2.7.2. Diastolic LV Dysfunction

Diastolic LV dysfunction was evaluated according to the 2009 ASE/EACVI recommendations and 2016 ASE/EACVI recommendations, and both were categorized in 4 grades: normal diastolic LV function and grade I–III (or intermediate) [25,26]. Regarding the 2016 ASE/EACVI recommendations, for patients with a preserved ejection fraction, four variables were evaluated: average mitral E/e’ velocity, septal and lateral e’ velocity, tricuspid regurgitation velocity (TR-velocity, cm/s) and LA volume index. Patients with at least three aberrant values were diagnosed with diastolic LV dysfunction. Patients with more than one missing variable were not classified.

#### 2.7.3. Hypertension

Patients were diagnosed with hypertension when they had a systolic blood pressure ≥ 140 mmHg or diastolic blood pressure ≥ 90 mmHg, measured during the physical examination, or when they were using antihypertensive medication.

#### 2.7.4. Family History

A positive family history for cardiovascular events or cerebrovascular events was defined as having a first-degree female relative under 65 years old or a male relative under 55 years old diagnosed with angina pectoris/myocardial infarction or TIA/CVA, respectively.

#### 2.7.5. Cardiovascular Diseases

A positive history of cardiovascular disease was defined as a history of angina pectoris, myocardial infarction, coronary artery bypass grafting (CABG), congestive heart failure, CVA/TIA and/or peripheral ischemia.

#### 2.7.6. Obesity

Obesity is defined as a BMI ≥ 30 kg/m^2^.

#### 2.7.7. Statistical Analysis

Continuous variables are expressed as mean ± standard deviation (SD) for normally distributed variables or median (interquartile range) for non-normally distributed variables. Differences were compared between subjects using the independent samples *t*-test for normally distributed variables and the Mann–Whitney U test for non-normally distributed variables. Dichotomous and categorical data are presented as frequencies (percentages). These variables were compared using a Chi-square test or Fisher’s exact test where applicable.

Logistic multivariate regression analyses were performed to investigate the association between AS and valve regurgitation. In addition, subgroup analyses using linear or logistic multivariate regression analyses were performed to investigate the relationship of disease duration, disease activity and anti-tumor necrosis factor (TNF) with valve regurgitation. All statistical analyses were performed using SPSS software (version 23.0, Chicago, IL, USA).

## 3. Results

### 3.1. Patient Characteristics

Baseline characteristics are shown in Table 1. A total of 193 AS patients and 74 matched controls were included in the study. Matching was performed in the best way possible within logistical limitations. The mean age of the AS patients and the controls was respectively 60 (±7) years and 62 (±7) years, and 72% and 58% were male, respectively. AS patients had a lower BMI and BSA than controls, respectively 26.6 ± 4.1 vs. 28.6 ± 5.5 kg/m^2^ and 1.9 ± 0.20 vs. 2.0 ± 0.2 m^2^. Furthermore, obesity and hypercholesterolemia were seen less often in AS patients compared to controls, respectively 22% vs. 30% and 19% vs. 30%. AS patients used more antihypertensives compared to controls, respectively 44% vs. 27%. The prevalence of other cardiovascular diseases, comorbidities and risk factors were comparable in both groups.

### 3.2. Disease Characteristics

Disease characteristics are shown in Table 2. A total of 82% of the AS patients were HLA-B27+. AS patients had a moderate to high disease activity with a mean ASDAS-CRP of 2.1 (±1.0) and a mean disease duration of 22 (±12) years. Most of the OA controls had knee osteoarthritis (82%) and/or poly-osteoarthritis (81%) compared to hip osteoarthritis (24%). Disease duration (since diagnosis) was 5 (2–8) years. OA patients had a median WOMAC score of 42 (±23).

### 3.3. Electrocardiography

Electrocardiographic results are shown in Table 3. No differences were found in conduction disorders between AS patients and controls.

### 3.4. Echocardiography

Table 4 provides an overview of echocardiographic parameters. An increased aortic root index was seen in AS patients compared to controls, though both in the normal range, respectively 1.74 (± 0.20) cm/m^2^ vs. 1.68 (± 0.22) cm/m^2^, *p* = 0.08. The prevalence of aortic root dilatation (≥2.1 cm/m^2^) was comparable in both groups, with a prevalence of 7% in AS patients and 4% in controls, *p* = 0.53. Furthermore, AS patients had AVR more often compared to controls, 41 (23%) vs. 8 (11%), *p* = 0.04. No difference was observed in MVR between AS patients and controls. The prevalence of systolic and diastolic LV dysfunction (both 2009 and 2016 ESE/EACVI grading criteria) was low and comparable between AS patients and controls.

### 3.5. Results of Regression Analyses

Results of logistic multivariate regression analyses are presented in Table 5. The relationship between AVR and AS was statistically significant and became more pronounced after adjusting for age, sex and CV risk factors. It revealed that AS patients had an increased odds of AVR compared to controls (OR: 4.5; 95% CI: 1.1–13.6). In contrast, AS patients did not have an increased odds of MVR compared to controls (OR: 1.9; 95% CI: 0.8–4.5). Lastly, subgroup analyses in AS showed no significant relationship between disease duration, ASDAS-CRP and anti-TNF drugs and the presence of AVR or MVR.

## 4. Discussion

The results of this study revealed that diastolic LV dysfunction, based on the 2016 criteria of the ASE/EACVI, was rare in both AS patients and controls, and the prevalence was comparable for both groups. Equally, the prevalence of conduction disorders was similar in both groups and mostly not clinically relevant. In contrast, AS patients had an up to five times increased odds of AVR compared to OA controls. Disease activity, disease duration and use of TNF-inhibitors were not associated with AVR.

### 4.1. Diastolic LV Dysfunction

It is hypothesized that development of diastolic LV dysfunction may result from coronary endothelium activation caused by systemic inflammation [27]. The activated endothelial cells cause cardiomyocytes to hypertrophy and stiffen, but also enable monocytes to enter cardiac tissue, where they trigger collagen production. The combined effect of these processes may lead to microvascular rarefaction, interstitial fibrosis and stiff cardiomyocytes, which impairs relaxation of the ventricles and thus induces diastolic LV dysfunction [27,28].

The current literature supports evidence for an increased risk of diastolic LV dysfunction in AS patients, albeit that varying prevalence rates have been reported (9–45%) [14,15]. Three studies assessed diastolic LV dysfunction using the combined set of parameters as recommended by the ASE/EACVI in 2009, ranging from 12% to 45%, though cut-off values of single parameters differed slightly between studies [15,29,30]. Other studies reported prevalence rates ranging from 20% to 49%, but those studies were not specifically designed to assess diastolic LV dysfunction or only used a single or few echocardiographic parameters for assessing diastolic LV dysfunction, which differs from the ASE/EACVI guidelines [8,14,26,31,32,33,34]. Nowadays, the most appropriate way, recommended by American and European echocardiography organizations (ASE and EACVI), to assess diastolic LV dysfunction, is to combine specific echocardiographic parameters [35].

However, the 2009 algorithms were considered too complex and had a substantial inter-observer variability, which possibly caused the wide variation in the observed prevalence of diastolic LV dysfunction. Therefore, the guidelines to assess diastolic LV dysfunction were upgraded in 2016 by the ASE/EACVI with the purpose of simplifying the approach [36]. It has been shown that the 2016 algorithm is superior to the 2009 algorithm with regards to specificity, correlation with clinical outcomes and inter-observer variability, but had a lower sensitivity [37]. Thus far, there have been no studies assessing diastolic LV function in AS patients with the updated ASE/EACVI 2016 guidelines. The above-mentioned studies mostly used the 2009 criteria, and when using the 2009 algorithm, we found a prevalence of diastolic LV dysfunction of 53% in AS patients and 46% in OA patients, respectively. When applying the 2016 criteria, these prevalence rates declined to 3.8% and 3.3%, respectively. Moreover, eight of the nine patients with diastolic LV dysfunction also had systolic LV dysfunction. According to the 2016 criteria, all patients with systolic LV dysfunction are defined to have also diastolic LV dysfunction. For our study, this means that only one patient in the entire cohort was diagnosed with diastolic LV dysfunction because of aberrant echocardiographic Doppler values. Altogether, our results indicate that diastolic LV dysfunction in AS patients is infrequent and that previous studies overestimated the prevalence of impaired diastolic LV function in AS patients due to the low accuracy of the diagnostic/grading tool.

### 4.2. Conduction Disorders

Major electrical conduction elements, such as the atrial-ventricular (AV) node and the bundle branches (BBs), are located in very close proximity to the heart valves. In addition to the aortic root and the cusps of the aortic valve, in AS, the inflammatory process therefore may extend to the atrial ventricular node (AV-node) and interventricular septum, leading to AR, AV-blocks and bundle branch blocks (BBB’s) [38]. However, most of the existing studies regarding conduction disturbances in AS patients were relatively small, and some lack controls and the results are inconsistent [8,11,12,39,40]. Our study assessed the clinically relevant and significant conduction disturbances in a large cohort of AS patients. We found a very low prevalence of, mostly mild, conduction disturbances with limited clinical relevance in the AS population comparable to the controls, which is in contrast to the existing literature [11,12,39]. A Swedish prospective, nationwide populations-based cohort showed that AS patients have a two-fold increased risk to develop an AV-block. However, the clinical relevance of this result is limited as this corresponds with an AV-block prevalence of 0.5% in AS patients compared to 0.4% in healthy subjects after a follow-up duration of 6 years [11]. Two other studies performed by Forsblad-d’Elia et al. and Dik et al. demonstrated in clinical trials that first-degree AV-blocks and BBB’s are common in AS patients [12,39]. However, both studies had no control group to compare their results to, and after specifying the type of BBB, the total prevalence of BBB’s dropped under 10%, with both studies reporting primarily incomplete BBB’s. In fact, Dik et al. demonstrated only 0.8% (1) of the AS patients with a complete right BBB, and none with a complete left BBB or left hemi block [39], though, their subjects were younger. Therefore, based on our results and in the context of the existing literature, we conclude that conduction disturbances are mostly mild with limited clinical relevance, and conduction disorders do not occur more frequently in AS patients compared to controls.

### 4.3. Aortic Valve Regurgitation

We found a prevalence of 23% of AVR in AS patients and thus a 4–5 times increased odds for AVR in comparison to controls. Previous echocardiographic studies showed varying prevalence rates of AVR in AS patients, ranging from 6% to 31% [6,8,9,10,38,40]. However, as the presence of AVR is associated with age and cardiovascular risk factors, and these differed greatly between the available studies, these prevalence data are not very comparable. Overall, the AS patients of these cohort studies were younger, had a shorter disease duration and one study did not include trace AVR, thus likely underestimating the prevalence of AVR. Brunner et al. observed that AVR was present in 10% of AS patients, but they included patients aged 32–86 years and did not assess CV risk factors or comorbidities [8]. Again, trace regurgitation was not taken into account, likely leading to a considerable underestimation of the prevalence.

Our most important observation was that elderly AS patients (50–75 years) have an up to five times increased odds of having AVR compared to controls after correction for age, sex and CV risk factors. In addition, OA patients have an increased CV burden due to increased CV comorbidities, and the increased risk that we observed in AS patients is probably an underestimation compared to when healthy subjects would have been used as controls [22].

In this study, we observed a significant difference in the prevalence of AV regurgitation between AS patients and controls, but manifestations were mostly mild. Nevertheless, chronic AVR is a slow progressive disease, and the chronic state of inflammation that is present in AS patients may further accelerate progression. Moreover, the American Heart Association/American College of Cardiology (AHA/ACC) and the European Association of Cardiovascular Imaging (EACVI) state that regurgitation of the aortic valve is pathological, regardless of its grade [41,42]. Hence, echocardiographic follow-up once per 3–5 years for trace or mild regurgitation, once per 1–2 years for moderate regurgitation and once per 0.5–1 year for severe regurgitation is recommended. Severe AR and rapid progression of the disease can be treated by valve replacement if recognized timely.

The current hypothesis on the valve involvement in AS is that entheses, the region where ligaments and tendons attach to bones, are the structures where inflammatory processes in AS mainly take place [43]. Pro-inflammatory cytokines, such as Interleukin (IL)- 23 and IL-17, have an important role in this inflammatory process, and IL-23 stimulates IL-17 production by Th17 cells, which further amplifies this inflammation [44,45,46]. The relevance for cardiac involvement in AS is that entheses and the part of the aortic valve that inserts into the aortic root are histologically similar [46]. Sherlock et al. demonstrated in mice that both entheses and this part of the aortic root contain IL-23 receptor-positive T-cells that can induce local inflammation after systemic exposure to IL-23 [46]. In the aortic root, inflammation may cause root dilatation and the inflammation may extend to the annulus, resulting in basal thickening and downward retraction of the cusps, also resulting in AVR [38,47,48]. The thickening of the annulus itself could also disturb the laminar blood flow, resulting in deterioration of valve function. In line with the increased prevalence of AVR in AS patients, we observed a trend towards a significantly greater aortic root index compared to OA controls, respectively 1.74 ± 0.20 cm/m^2^ vs. 1.68 ± 0.22 cm/m^2^, *p* = 0.08. This is consistent with small-sized studies of Roldan et al. and Yildirir et al., as they also showed increased aortic root diameter, as well as increased prevalence of AVR in AS patients compared to controls [38,49].

### 4.4. Strengths and Limitations

Our study has several strengths and limitations. First, to our knowledge, this is the largest study undertaken in AS patients assessing LV function by echocardiography. Second, this is the first study to assess diastolic LV dysfunction in AS patients based on the 2016 guidelines of the ASE/EACVI.

There are also limitations of the present study. First of all, due to the cross-sectional study design, the associations found in this study are not necessarily causal. We were therefore unable to determine long-term consequences of the cardiac manifestations we observed in our patients. Secondly, complete matching of groups on a ratio of 2:1 based on age, sex and current smoking status was not completely achieved, introducing minor differences in patient characteristics. Therefore, we adjusted for these variables in our regression analyses, thereby limiting the consequences thereof.

## 5. Conclusions

Against our expectations, the prevalence rates of diastolic LV dysfunction and conduction disorders were mostly not clinically relevant and similar in AS patients and controls. In contrast, AS patients have an up to five times increased odds of AVR, although this was mostly mild. However, it is important to realize that any stage of AVR is considered to be pathological as mild regurgitation may progress and result into severe complications. When timely recognized, it can be treated adequately (aortic valve replacement). Therefore, our findings indicate that echocardiographic screening of elderly AS patients (50–75 years) should be considered. Obviously, prospective studies should assess the cost-effectiveness of screening of all AS patients as well as the long-term complications of AVR in AS patients.

## Figures and Tables

**Table 1 jcm-10-05069-t001:** Patient characteristics of AS and OA controls.

	AS Patients (*n* = 193)	OA Controls (*n* = 74)
**Patient characteristics**		
Gender, male (*n*, %)	138 (72)	43 (58)
Age, years (mean ± SD)	60 ± 7	62 ± 7
Race, Caucasian (*n*, %)	162 (84)	65 (88)
BMI, kg/m^2^ (mean ± SD)	26.6 ± 4.1	28.6 ± 5.5
BSA, m^2^ (mean ± SD)	1.9 ± 0.2	2.0 ± 0.2
Blood pressure, mmHg		
Systolic (mean ± SD)	134 ± 16	133 ± 17
Diastolic (mean ± SD)	84 ± 8	82 ± 9
**CVD risk factors**		
Smoking status		
current (*n*, %)	39 (20)	15 (20)
ever (*n*, %)	98 (51)	35 (47)
never (*n*, %)	55 (29)	24 (32)
pack years (median, IQR)	28 (14–37)	12 (6–31)
Obesity (*n*, %)	42 (22)	26 (35)
Hypertension (*n*, %)	118 (61)	45 (61)
Hypercholesterolemia (*n*, %)	36 (19)	22 (30)
Diabetes Mellitus type II (*n*, %)	22 (11)	10 (14)
Total history CVD (*n*, %)	21 (11)	8 (11)
Angina Pectoris (*n*, %)	5 (3)	2 (3)
Myocardial infarction (*n*, %)	12 (6)	4 (5)
Stroke		
TIA (*n*, %)	4 (2)	3 (4)
CVA (*n*, %)	3 (2)	0 (0)
Peripheral ischemia (*n*, %)	1 (1)	1 (1)
CABG (*n*, %)	8 (4)	2 (3)
Family history (first degree)		
AP/MI (*n*, %)	24 (12)	11 (15)
Stroke/TIA (*n*, %)	14 (7)	6 (8)
**Laboratory**		
LDL, mmol/L	3.3 ± 0.9	3.5 ± 1.1
HDL, mmol/L	1.5 ± 0.4	1.3 ± 0.4
Cholesterol/HDL ratio	3.8 ± 1.6	4.2 ± 1.7
ESR, mm/h	7.5 (4.0–10.0)	5.0 (2.0–11)
CRP, mg/L	2.8 (1.1–7.8)	1.6 (0.8–3.1)
**Medication**		
Antihypertensives (*n*, %)	85 (44)	20 (27)
Lipid-modifying drugs (*n*, %)	40 (21)	19 (26)
NSAIDs (*n*, %)	108 (56)	37 (50)
Anti-TNF drugs		
current (*n*, %)	70 (36)	N.A.
ever (*n*, %)	13 (7)	N.A.
naive (*n*, %)	110 (57)	N.A.

Values are displayed as mean ± standard deviation (SD), median with corresponding interquartile range (IQR) or frequencies with corresponding percentages (%). AS = ankylosing spondylitis, OA = osteoarthritis, BMI = body mass index, BSA = body surface area, CVD = cardiovascular disease, TIA = transient ischemic attack, CVA = cerebrovascular attack, CABG = coronary artery bypass graft, AP = angina pectoris, MI = myocardial infarction, NSAIDs = non-steroidal anti-inflammatory drugs, anti-TNF = anti-tumor necrosis factor, N.A. = not applicable. Normal ranges for laboratory assessments: LDL 2.0–4.5 mmol/L, HDL 0.9–1.7 mmol/L, ESR in men > 50 years < 20 mm/h, ESR in women > 50 years < 30 mm/h, CRP < 10 mg/L.

**Table 2 jcm-10-05069-t002:** Disease characteristics of AS and OA controls.

Ankylosing Spondylitis	*n* = 193
HLA-B27-positive (*n*, %)	156 (82)
Disease activity and severity	
ASDAS-CRP (mean ± SD)	2.1 ± 1.0
BASMI (mean ± SD)	4.1 ± 1.8
BASFI (mean ± SD)	3.7 ± 2.4
BASDAI (median; IQR)	3.1 (1.6–5.0)
Time since diagnosis, years (mean ± SD)	22 ± 12
Osteoarthritis	*n* = 74
Affected joints	
Knee (*n*, %)	61 (82)
Hip (*n*, %)	18 (24)
Poly-osteoarthritis (*n*, %)	60 (81)
Prosthesis (*n*, %)	15 (20)
WOMAC (mean ± SD)	42 (23)
Time since diagnosis (median; IQR)	5 (2–8)

Values are displayed as mean ± standard deviation (SD), median with corresponding interquartile range (IQR) or frequencies with corresponding percentages (%). AS = ankylosing spondylitis, OA = osteoarthritis, HLA = Human leukocyte antigen, ASDAS-CRP = Ankylosing Spondylitis Disease Activity Score-C-reactive protein, BASMI = Bath Ankylosing Spondylitis Metrology Index, BASFI = Bath Ankylosing Spondylitis Functional Index, BASDAI = Bath Ankylosing Spondylitis Disease Activity Index, WOMAC = Western Ontario and McMaster Universities Osteoarthritis.

**Table 3 jcm-10-05069-t003:** Electrocardiographic results.

	AS Patients (*n* = 193)	OA Controls (*n* = 74)	*p*-Value ^a^
Atrial fibrillation (*n*, %)	3 (2)	4 (5)	0.10
Atrial flutter (*n*, %)	0 (0)	0 (0)	1.0
AV-block			
1st degree (*n*, %)	2 (1)	1 (1)	0.62
2nd degree, Mobitz type 1 (*n*, %)	0 (0)	0 (0)	1.0
2nd degree, Mobitz type 2 (*n*, %)	0 (0)	0 (0)	1.0
3rd degree (*n*, %)	0 (0)	0 (0)	1.0
LBBB (*n*, %)	2 (1)	0 (0)	1.0
LAFB (*n*, %)	2 (1)	0 (0)	1.0
RBBB (*n*, %)	2 (1)	3 (4)	1.0
iRBBB (*n*, %)	13 (7)	5 (7)	1.0
Pathologic Q waves (*n*, %)	3 (2)	4 (5)	0.10
LVH (*n*, %)	9 (5)	0 (0)	0.11
Nonspecific IVCD (*n*, %)	2 (1)	1 (1)	1.0
Pacemaker (*n*, %)	2 (1)	0 (0)	1.0
Other (*n*, %)	5 (3)	0 (0)	0.33
Total (*n*, %)	44 (23)	18 (24)	

Values are displayed as frequencies with corresponding percentages (%). AS = ankylosing spondylitis, OA = osteoarthritis, AV block = atrioventricular block, LBBB = left bundle branch block, LAFB = left anterior fascicular block, RBBB = right bundle branch block, iRBBB = incomplete right bundle branch block, LVH = left ventricular hypertrophy, IVCD = intraventricular conduction delay. ^a^ *p*-values of chi-square test.

**Table 4 jcm-10-05069-t004:** Echocardiographic results.

Cardiac Structures	AS Patients (*n* = 193)	OA Controls (*n* = 74)	*p*-Value ^a^
Aortic root index, cm/m^2^ (mean ± SD)	1.74 ± 0.20	1.68 ± 0.22	0.08
Aortic root dilatation, ≥2.1 cm/m^2^ (*n*, %)	12 (7)	2 (4)	0.53
** Mitral valve regurgitation			0.46
Mild (*n*, %)	65 (34)	23 (32)	
Moderate (*n*, %)	4 (2)	0 (0)	
Severe (*n*, %)	0 (0)	0 (0)	
Prosthesis (*n*, %)	0 (0)	0 (0)	
Aortic valve regurgitation			0.04 *
Trace (*n*, %)	16 (9)	1 (1)	
Mild (*n*, %)	23 (12)	7 (10)	
Moderate (*n*, %)	1 (1)	0 (0)	
Severe (*n*, %)	1 (1)	0 (0)	
Prosthesis (*n*, %)	1 (1)	0 (0)	
Cardiac function			
LV mass index, g/m^2^ (mean ± SD)	75 ± 20	76 ± 19	0.79
LA volume index, mL/m^2^ (mean ± SD)	29 ± 9	32 ± 13	0.11
EDV index, mL/m^2^ (mean ± SD)	62 ± 15	55 ± 18	0.01 *
ESV index, mL/m^2^ (mean ± SD)	27 ± 8	22 ± 10	<0.01 *
Ejection fraction (mean ± SD)	57 ± 6	60 ± 8	0.02 *
E/e’ average, cm/s (mean ± SD)	8.5 ± 2.5	8.0 ± 2.1	0.16
E-max, cm/s (mean ± SD)	69 ± 17	65 ± 17	0.06
A-max, cm/s (mean ± SD)	71 ± 17	71 ± 17	0.95
E/A ratio (mean ± SD)	1.0 ± 0.3	0.9 ± 0.2	0.03 *
MV deceleration time, m/s (mean ± SD)	0.22 ± 0.05	0.22 ± 0.04	0.83
Septal e’ velocity, cm/s (mean ± SD)	7.8 ± 1.9	7.4 ± 1.7	0.12
Lateral e’ velocity, cm/s (mean ± SD)	9.5 ± 2.7	9.5 ± 2.2	0.94
TR velocity, cm/s (mean ± SD)	219 ± 26	229 ± 30	0.23
Systolic LV dysfunction (*n*, %)	10 (5)	2 (3)	0.74
Diastolic LV dysfunction—2016			0.88
Grade I (*n*, %)	6 (3)	2 (3)	
Grade II (*n*, %)	1 (1)	0 (0)	
Grade III (*n*, %)	0 (0)	0 (0)	
Diastolic LV dysfunction—2009			0.59
Grade I (*n*, %)	60 (32)	17 (25)	
Grade II (*n*, %)	39 (21)	16 (24)	
Grade III (*n*, %)	0 (0)	0 (0)	

Values are displayed as mean ± standard deviation (SD) or frequencies with corresponding percentages (%). Indexed values are corrected for BSA. AS = ankylosing spondylitis, OA = osteoarthritis, BSA = body surface index, LVMI = left ventricle mass index, LV = left ventricular, EDV = end diastolic volume, ESV = end systolic volume, MV = mitral valve, TR = tricuspid regurgitation. ^a^ *p*-values of Student’s *t*-test or chi-square test. * Bold, significance level of *p* ≤ 0.05. ** A trace MVR is considered as physiological regurgitation and is therefore not added in the table.

**Table 5 jcm-10-05069-t005:** Association of AS with valve regurgitation.

	Crude Model	Adjusted Model ^a^	Fully Adjusted Model ^b^
Aortic valve regurgitation	2.3 (1.0–5.3) *	2.9 (1.3–6.8) *	4.5 (1.1–13.6) *
Mitral valve regurgitation	1.2 (0.7–2.2)	1.4 (0.8–2.5)	1.9 (0.8–4.5)

Values are displayed as odds ratio with corresponding 95% confidence interval. ^a^ Adjusted for age and gender. ^b^ Adjusted for age, gender, current smoking status, pack years, systolic blood pressure, diastolic blood pressure, hypercholesterolemia, BMI, Diabetes Mellitus type 2, family history of cardiovascular diseases. * Significance level of *p* ≤ 0.05.

## Data Availability

The datasets generated during and/or analyzed during the current study are available from the corresponding author upon reasonable request.

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
