# Peer review of "The Prevalence of Cardiac Diseases in a Contemporary Large Cohort of Dutch Elderly Ankylosing Spondylitis Patients—The CARDAS Study"

_jcm, 2021, doi:10.3390/jcm10215069_

Round 1

Reviewer 1 Report

It was a cross-sectional controlled Study in a large cohort with matched controls and the cardiac examinations were perfectly described. We need to update the data on the prevalence of extra-articular manifestations in patients with AD in the era of biological treatments. Older series may not coincide with recent cohorts, in which the duration of inflammation has been better controlled and therefore organ damage is likely to have been reduced. It is not original because other authors have addressed this issue but in small series and without a control group. 

An update of the data on the risk of aortic valve regurgitation in patients with ankylosing spondylitis and the result that the prevalence of left ventricular diastolic dysfunction and conduction disorders were not relevant.

I think it is well written, I understood every part of the article. And, it is well written and easy to read. The conclusions are in line with the study and the results found.

Author Response

Author Response 1

Response 1: We thank the reviewer for the positive comments.

Reviewer 2 Report

Manuscript ID  jcm-1440352

"The prevalence of cardiac diseases in a contemporary large cohort of Dutch ankylosing spondylitis patients - the CARDAS study”

A study evaluating both echocardiographic and electrocardiographic changes in AS patients.

Minor recommendations: Why was the 50-75 age range chosen specifically?

Patients with OA has selected as the control group. There are differences in both gender and body mass index.

In Table 1, the characteristics of both groups are given, but there are no p values. Both groups should be compared with appropriate statistical methods and the p values should be added to Table 1.

Normal ranges of laboratory parameters should be given.

Author Response

"The prevalence of cardiac diseases in a contemporary large cohort of Dutch ankylosing spondylitis patients - the CARDAS study”

 A study evaluating both echocardiographic and electrocardiographic changes in AS patients.

Point 1: Minor recommendations: Why was the 50-75 age range chosen specifically?

Response 1: This study additionally aimed to assess whether echocardiographic/electrocardiographic screening is of added clinical value. Therefore, as cardiac disease is more prominent in elderly people and the medical benefit of cardiac screening in very old patients (>75 years) is doubtful we chose to include AS patients between 50 and 75 years

Point 2: Patients with OA has selected as the control group. There are differences in both gender and body mass index. In Table 1, the characteristics of both groups are given, but there are no p values. Both groups should be compared with appropriate statistical methods and the p values should be added to Table 1.

Response 1: We thank the reviewer for this comment. However,  we chose not to present p-values in the baseline table, where we present the patients characteristics of two different cohorts. P-values in this table would present the probability of whether the differences in the variables are based on chance. However, as these groups are totally different any baseline difference which is observed (beyond age, sex and smoking as these were matched) is by definition due to chance, and therefore, p-value testing is illogical. Differences in the patients characteristics should be noticed and placed in context regardless of a p-value. Therefore, in our opinion and in line with the literature1,2,3, we felt that a p-value is not of added value in baseline tables presenting patients characteristics of two different cohorts.

  • http://www.consort-statement.org/checklists/view/32--consort-2010/510-baseline-data
  • Knol M, Groenwold R, Grobbee D. P-values in baseline tables of randomised controlled trials are inappropriate but still common in high impact journals. European Journal of Preventive Cardiology. 2012;19(2):231-232. doi:10.1177/1741826711421688
  • Palesch YY. Some common misperceptions about P values. 2014;45(12):e244-e246. doi:10.1161/STROKEAHA.114.006138

Point 3: Normal ranges of laboratory parameters should be given.

Response 1: We have added the normal ranges of laboratory parameters accordingly. Page 5 , line 179-180.

Normal ranges for laboratory assessments: LDL 2.0 - 4.5 mmol/l, HDL 0.9 – 1.7 mmol/l, ESR in men >50 years <20mm/hr, ESR in women >50 years <30mm/hr, CRP <10mg/l…

Furthermore we added the footnote of table 1 as this was missing and we added the word “elderly” in the title as this gives a better description of the presented data in this manuscript.